# Comparing Antibody Responses to Homologous vs. Heterologous COVID-19 Vaccination: A Cross-Sectional Analysis in an Urban Bangladeshi Population

**DOI:** 10.3390/vaccines13010067

**Published:** 2025-01-13

**Authors:** Kazi Istiaque Sanin, Mansura Khanam, Azizur Rahman Sharaque, Mahbub Elahi, Bharati Rani Roy, Md. Khaledul Hasan, Goutam Kumar Dutta, Abir Dutta, Md. Nazmul Islam, Md. Safiqul Islam, Md. Nasir Ahmed Khan, Mustufa Mahmud, Nuzhat Nadia, Fablina Noushin, Anjan Kumar Roy, Protim Sarker, Fahmida Tofail

**Affiliations:** 1International Centre for Diarrhoeal Disease Research, Bangladesh, 68, Shaheed Tajuddin Ahmed Sarani, Dhaka 1212, Bangladesh; mansura@icddrb.org (M.K.); azizurrahmanmars@gmail.com (A.R.S.); mahbub.elahi@icddrb.org (M.E.); bharati.roy@icddrb.org (B.R.R.); khaledul.hasan@icddrb.org (M.K.H.); goutam.dutta@icddrb.org (G.K.D.); abir.dutta@icddrb.org (A.D.); anjan@icddrb.org (A.K.R.); protim@icddrb.org (P.S.); ftofail@icddrb.org (F.T.); 2Communicable Disease Control (CDC), Directorate General of Health Services, Ministry of Health and Family Welfare, Dhaka 1212, Bangladesh; nimunna@yahoo.com (M.N.I.); 67shafikislam@gmail.com (M.S.I.); nasir_7@hotmail.com (M.N.A.K.); dr.mustufa@gmail.com (M.M.); dr.mrittika07@gmail.com (N.N.); fablinanoushin17@gmail.com (F.N.)

**Keywords:** COVID-19, homologous and heterologous vaccination, COVID-19 booster vaccinations, Bangladesh

## Abstract

Background: Vaccination has played a crucial role in mitigating the spread of COVID-19 and reducing its severe outcomes. While over 90% of Bangladesh’s population has received at least one COVID-19 vaccine dose, the comparative effectiveness of homologous versus heterologous booster strategies, along with the complex interplay of factors within the population, remains understudied. This study aimed to compare antibody responses between these booster approaches. Methods: This cross-sectional study enrolled 723 adults in urban Dhaka who had received COVID-19 booster doses within the last six months. Participants were grouped based on homologous or heterologous booster vaccination. Data were collected through structured household surveys, and 2 mL blood samples were collected for measuring antibody titers. Results: Heterologous booster recipients showed higher median antibody titers (8597.0 U/mL, IQR 5053.0–15,482.3) compared to homologous recipients (6958.0 U/mL, IQR 3974.0–12,728.5). In the adjusted analysis, the type of booster dose had no significant impact on antibody levels. However, the duration since the last booster dose was significantly associated with antibody levels, where each additional month since receiving the booster corresponded to approximately a 15–16% reduction in antibody levels (Adj. coeff: 0.85, 95% CI: 0.81, 0.88; *p* < 0.001). Participants over 40 years demonstrated higher antibody levels than younger individuals (Adj. coeff: 1.23, 95% CI: 1.07, 1.43; *p* = 0.005). Sex, BMI, and prior COVID-19 infection showed no significant associations with antibody levels after adjustment. Conclusion: The results underscore the complexity of immune responses across different demographic groups and suggest potential benefits of ongoing heterologous booster strategies in sustaining immunity.

## 1. Introduction

The COVID-19 pandemic, caused by the highly contagious Severe Acute Respiratory Syndrome Coronavirus 2 (SARS-CoV-2) virus [1], has profoundly impacted global public health since its emergence in late 2019 [2]. Vaccination has been central to controlling the virus’s spread, with millions receiving initial doses aimed at reducing severe illness, hospitalizations, and mortality rates [3,4]. High-income nations, including the US and Europe, prioritized mRNA vaccines such as Pfizer-BioNTech and Moderna in their immunization programs [5]. In contrast, South Asian countries focused on inactivated vaccines like Covaxin and adenovirus-based vaccines like Covishield, driven by considerations of production capacity and affordability. However, the continuous emergence of new variants [6] and the observed decline in immunity over time have made booster vaccinations an essential strategy to sustain protection [4,7,8,9]. This is particularly crucial in densely populated regions with diverse sociodemographic profiles, such as Bangladesh.

Bangladesh’s vaccination campaign commenced with the Oxford-AstraZeneca (Adenovirus vector vaccine) and later included mRNA vaccines like Pfizer-BioNTech (BNT162b2) and Moderna (mRNA-1273), which have been instrumental in mitigating the pandemic’s impact [10,11,12]. Specifically, Dhaka, the capital of Bangladesh, has received the highest number of COVID-19 vaccine doses, largely facilitated by the COVAX initiative. To date, over 150 million people in Bangladesh have received the first dose, 142 million the second dose, and 68 million the third or booster dose. Overall, more than 90% of the population in Bangladesh has received at least one dose of the COVID-19 vaccine [13]. Despite these efforts, the decline in immunity following initial vaccination and the emergence of SARS-CoV-2 variants capable of evading immune responses have highlighted the critical need for booster doses [14]. Current evidence suggests that immunity provided by the primary vaccination wanes within 4–6 months, making it essential to administer a booster when available [14,15,16].

In this context, both homologous and heterologous vaccine regimens have been explored to optimize immune responses and ensure broad vaccine coverage. A homologous regimen, which uses the same vaccine for both the primary series and the booster dose, has been the standard approach. However, the inconsistent vaccine supply and procurement challenges in Bangladesh [17] have led to situations where certain vaccine types were unavailable. As a result, heterologous regimens using a different vaccine for the booster dose compared to the primary series have emerged as a practical alternative [18,19]. This approach not only provides greater flexibility in vaccine distribution but has also been shown in some studies to enhance immunogenicity. Research suggests that heterologous booster strategies may offer stronger immune response, making them a valuable option in settings with inconsistent vaccine supply [20,21].

Despite the widespread implementation of booster programs in Bangladesh, significant gaps remain in the data regarding the effectiveness of these strategies within the population. Additionally, the complex interplay of factors such as age, sex, and nutritional status with homologous and heterologous vaccine types is not well understood. Given Bangladesh’s unique, densely populated sociodemographic landscape, these factors could substantially influence vaccination outcomes. This study aims to address these critical knowledge gaps by estimating and comparing antibody responses to homologous and heterologous COVID-19 booster vaccinations among recipients in Dhaka, Bangladesh. Our research question was as follows: ‘What are the differences in antibody responses between homologous and heterologous COVID-19 booster vaccinations among recipients in Dhaka, Bangladesh, and how do other factors influence these outcomes?’.

## 2. Materials and Methods

### 2.1. Study Design

This cross-sectional study was designed to estimate and compare the antibody responses to homologous and heterologous COVID-19 booster vaccinations among recipients in Dhaka, Bangladesh.

### 2.2. Study Population

This study included 723 adults aged 18 and above who received a Pfizer or Moderna COVID-19 booster dose within the last six months. Participants were randomly selected from areas in the Dhaka division where vaccination rates are higher, including Mirpur, Mohammadpur, and Savar, using the Directorate General of Health Services database. Eligible participants had completed their two-dose primary vaccination series and received either a homologous or heterologous booster.

The sample size was calculated using a 95% confidence level (Z = 1.96), an assumed prevalence of 50% (*p* = 0.5) due to the lack of prior studies, and a precision limit of 5% (D = 0.05). Based on this, the target was 800 participants, ensuring sufficient power to compare booster groups. Participants were contacted via phone, and written consent was obtained before enrollment. Individuals with severe illness or contraindications to blood sampling were excluded, resulting in a final sample of 723 participants.

### 2.3. Study Setting

Data collection took place in selected areas of the Dhaka division, Bangladesh, with field teams visiting participants’ homes to gather data and biological samples. This setting was chosen due to high vaccination coverage, the availability of participants who met the study criteria, and logistical support aligned with the study budget. Data collection occurred between November 2022 and June 2023.

### 2.4. Data Collection

Data were collected through structured household surveys and blood sampling. The survey collected information on sociodemographic characteristics, medical history, and psychosocial factors. The questionnaires were developed in English, translated into Bengali, and pre-tested for relevance and clarity. Certified phlebotomists collected 2 mL blood samples following strict protocols to minimize variability. The samples were immediately labeled, centrifuged in the nearby laboratory settings and transported in temperature-controlled cooler boxes to the Immunobiology, Nutrition, and Toxicology Laboratory (INTL) at icddr,b, where they were processed and stored at −70 °C until analysis. SARS-CoV-2-specific antibodies were measured quantitatively on a fully automated immunoassay analyzer, Cobas e601 (Roche Diagnostics GmbH, Mannheim, Germany), based on an electro-chemiluminescence immunoassay in a double-antigen sandwich assay format using Elecsys Anti-SARS-CoV-2 S. This assay quantified total antibodies (IgG/IgA/IgM) targeting the SARS-CoV-2 S-RBD protein, with PreciControl Anti-SARS-CoV-2 S used to monitor accuracy.

### 2.5. Study Definitions

#### 2.5.1. Homologous Booster Vaccination

In this study, a homologous booster vaccination refers to the administration of a booster dose using the same vaccine brand as the initial two-dose primary series. For instance, participants who received two doses of the Pfizer-BioNTech vaccine followed by a Pfizer-BioNTech booster are classified as having received a homologous booster.

#### 2.5.2. Heterologous Booster Vaccination

A heterologous booster vaccination refers to the administration of a booster dose with a different vaccine brand than that used for the initial two-dose primary series. For example, participants who received two doses of the AstraZeneca vaccine and then received a booster with either the Pfizer-BioNTech or Moderna vaccine are classified as having received a heterologous booster.

### 2.6. Assessment Tools and Variables

The primary outcome variable targeted was the level of serum antibodies against SARS-CoV-2 after booster vaccination, expressed in U/mL. Secondary variables included sociodemographic factors such as age, sex, education, and occupation, measured using structured questionnaires. The time since the last booster dose was recorded as a continuous variable. Nutritional status was assessed by measuring participants’ height and weight using a stadiometer and electronic weighing scale at household levels to calculate BMI. Health status was evaluated through questions about any chronic diseases. The Raven test was used to assess participants’ non-verbal intelligence and abstract reasoning. To measure five dimensions of mental well-being (usual activities, pain/discomfort, anxiety/depression), the EQ-5D-5L tool was applied, scoring each dimension at five levels: no problems, slight, moderate, severe, and extreme problems. Additionally, questions gathered information on participants’ mental, social, and physical well-being post-vaccination, covering anxiety, discomfort, forgetfulness, difficulties in daily activities, physical weaknesses, concentration, and appetite changes since the last vaccination.

### 2.7. Data Analysis

Descriptive statistics were used to summarize the participants’ sociodemographic characteristics, health status, and antibody levels. Bivariate associations were examined using the Wilcoxon rank sum test, Kruskal–Wallis rank sum test, and Spearman’s rank correlation to compare antibody responses between the homologous and heterologous groups, as well as sociodemographic characteristics and health statuses. We applied log transformation as the antibody titer did not meet the normality assumption. Simple and multiple linear regression analyses were conducted, adjusted for potential covariates, including age, sex, BMI, and history of COVID-19 infection. The results of these regression analyses are presented as exponential beta coefficients with 95% confidence intervals. All statistical analyses and graphical presentations were performed using R Studio version 4.2.1, with a *p*-value of <0.05 considered statistically significant.

## 3. Results

The study population included 723 participants, with 279 receiving homologous booster vaccinations and 444 receiving heterologous boosters. The mean age in the homologous group was higher (41.3 years, SD 13.50) compared to the heterologous group (36.8 years, SD 11.85). Additionally, a larger proportion of the homologous group were aged over 40 (52.33%) compared to the heterologous group (34.68%).

Educational attainment was slightly higher in the heterologous group, with 19.82% having higher education compared to 11.11% in the homologous group. A significantly larger proportion of participants in the heterologous group were engaged in service jobs (52.03%) compared to the homologous group (27.60%).

While the wealth index and BMI distributions were relatively similar between the groups, a higher percentage of participants in the heterologous group belonged to the “rich” wealth category (22.97%) compared to the homologous group (8.24%). The detailed results on the sociodemographic characteristics are provided in Table 1.

Overall, the data (Table 2) indicate that participants in the homologous group had a higher burden of chronic diseases compared to those in the heterologous group, particularly for conditions like diabetes and heart disease. Known diabetes was more prevalent in the homologous group (11.11%) than in the heterologous group (6.31%). Similarly, diagnosed heart disease was more common in the homologous group (5.38%) compared to the heterologous group (2.93%). Other chronic conditions showed similar prevalence rates between the two groups.

Most participants, based on self-reporting, had not been infected with COVID-19 or COVID-like symptoms, with 95.44% indicating no prior infection. However, a small percentage had experienced COVID-19 more than once, with slightly higher rates in the heterologous group (1.35%) compared to the homologous group (0.36%).

Functional health status varied between the groups, with a higher proportion of participants in the heterologous group reporting difficulties with self-care (45.05%) and usual work activities (49.10%) compared to the homologous group (36.92% and 37.28%, respectively). Pain was also more frequently reported in the heterologous group (47.30%) than in the homologous group (29.39%).

In terms of mental health indicators, anxiety and difficulty concentrating were slightly more common in the homologous group, while difficulties with memory were similar between both groups. Feelings of weakness were reported by approximately one-third of participants in both groups.

The median antibody titer levels were found to be higher in participants who received heterologous booster vaccinations compared to those who received homologous boosters. The median titer level for the heterologous group was 8597.00 U/mL, with an interquartile range (IQR) of 5053.00 to 15,482.25 U/mL. In contrast, the homologous group exhibited a median titer level of 6958.00 U/mL, with an IQR of 3974.00 to 12,728.50 U/mL (Table 1, Figure 1).

Figure 2 illustrates the trajectory of antibody responses over time in participants who received homologous and heterologous COVID-19 booster vaccinations. In both groups, antibody levels exhibited a decline over time following the time of booster dose. However, the magnitude and rate of decline differed between the two groups. In the heterologous group, initial antibody titers were notably higher compared to the homologous group. Despite the gradual decrease, the heterologous group presented higher antibody levels compared to their homologous counterparts when the duration after receiving a booster dose was considered.

Our analysis (Table 3) revealed that the type of vaccination had a notable impact on antibody levels, though the significance of this effect reduced after adjustment for other covariates. In the unadjusted analysis, participants who received heterologous boosters had 23% higher antibody levels compared to those who received homologous boosters (coefficient: 1.23, 95% CI: 1.08, 1.40; *p* = 0.001). However, after adjusting for other factors, this association was no longer statistically significant (coefficient: 1.10, 95% CI: 0.97, 1.25; *p* = 0.146).

The duration since the last booster dose was significantly associated with antibody levels, with a consistent decline in antibodies over time in both unadjusted and adjusted models. Each additional month since receiving the booster was associated with approximately a 15–16% reduction in antibody levels (adjusted coefficient: 0.85, 95% CI: 0.81, 0.88; *p* < 0.001).

Age also played a significant independent role, with participants over 40 years of age showing higher antibody levels compared to those under 30 years of age in the adjusted model (coefficient: 1.23, 95% CI: 1.07, 1.43; *p* = 0.005). This suggests that older participants, despite the general age-related decline in immune function, may have experienced a stronger antibody response following booster vaccination.

Sex differences were observed in the unadjusted analysis, where females had significantly lower antibody levels compared to males (coefficient: 0.80, 95% CI: 0.70, 0.90; *p* < 0.001). However, this association became non-significant after adjustment (*p* = 0.099). With regard to wealth index/BMI/having a history of COVID-19 infection, none of these were significantly associated with antibody levels after adjustment.

## 4. Discussion

This study aimed to evaluate and compare antibody responses to homologous and heterologous COVID-19 booster vaccinations among recipients in Dhaka, Bangladesh. Additionally, it explored how other factors influenced these immune responses. The findings offer valuable insights into the effectiveness of different booster strategies in a densely populated and sociodemographically diverse setting, providing critical information for public health planning in Bangladesh and similar regions.

Our findings indicate that the type of vaccination had a significant impact on antibody levels at the initial level, with the median antibody titer being higher in the heterologous booster group. These results are consistent with recent studies and meta-analyses, which have reported enhanced immune responses with heterologous booster strategies. Heterologous boosting has been shown to result in similar or greater increases in binding and neutralizing antibody titers compared to homologous boosting, providing a broader range of immune responses [21]. Overall, heterologous immunization generated significantly higher antibody responses 14 to 28 days post-booster compared to homologous immunization [22]. The potential advantages of heterologous boosting may stem from the engagement of diverse immune pathways by different vaccine platforms, leading to a more robust and broader immune response. However, the wide variability in antibody responses within both the homologous and heterologous group suggests that individual immune factors may play a more critical role than the vaccination strategy itself.

Despite the initially higher antibody levels in the heterologous group, our adjusted analysis showed that this difference diminished when controlling for time since vaccination. We observed a significant decline in antibody levels over time following booster vaccination, with a 15–16% reduction per month. This finding aligns with the well-documented waning of vaccine-induced immunity and emphasizes the importance of timing in booster strategies. Recent systematic reviews have noted a 21% decline in vaccine effectiveness within six months after full vaccination and a 29% reduction in protection against symptomatic infections 1–4 months after a booster during the Omicron period [8,23]. Other studies confirm that antibody levels drop three months after BNT162b2 vaccination, continuing to decline, with effective levels diminishing by six months [24,25,26].

Evidence suggests that the efficacy of mRNA vaccines, such as Pfizer-BioNTech and Moderna, decreases over time due to diminishing humoral immunity, with antibody levels peaking 21–28 days post-second dose and progressively decreasing 4–6 months later [27]. This decline appears to be time-dependent rather than influenced by patient factors. Consequently, a three-dose regimen, whether heterologous or homologous, is considered essential for preventing COVID-19 variant infections, with both approaches proving equally effective [28]. In our study, the rate of antibody decline was similar between the homologous and heterologous groups, suggesting that while heterologous boosting may initially induce higher antibody levels, its durability may not differ substantially from homologous boosting. This underscores the need for the continued monitoring of antibody levels and the consideration of additional booster doses, regardless of the initial strategy employed.

Contrary to common expectations, our study found that participants over 40 exhibited higher antibody levels than those under 30, even after adjusting for other factors. This challenges the typical assumption that immune function and vaccine responsiveness decline with age [29]. While most studies suggest that older individuals tend to have lower antibody titers [30,31,32,33,34], our results indicate a more complex relationship.

Reduced vaccine efficacy in the elderly is often attributed to adaptive immunosenescence, which includes diminished responses to new antigens, reduced memory T cell capacity, and chronic low-level inflammation, all of which contribute to weaker immune responses [35,36]. For example, Khoury J et al. reported lower antibody titers in participants over 50 compared to younger individuals, who showed significantly higher IgG and neutralizing antibody levels [37]. Similarly, a study from Japan observed significantly lower antibody levels in older participants, with much higher median titers in those in their 20s compared to individuals in their 60s and 70s [38]. Other research has consistently found stronger antibody responses in younger individuals after the second dose of BNT162b2 [39,40,41,42].

However, some studies have presented contrasting evidence. Wu et al. found higher neutralizing antibody levels in older patients compared to younger individuals [43], and Ozgocer, T et al. also observed stronger anti-SARS-CoV-2 responses in males and older individuals [44]. On the other hand, some research found no clear correlation between age and antibody concentrations [45,46].

The higher antibody levels observed in older participants in our study could be attributed to prior infections, which may have primed their immune systems for a stronger response to COVID-19 vaccination. However, in this study, the majority of respondents (~96%) reported no history of infection. We suspect that many were unable to recognize or confirm an infection due to the absence of proper diagnostic evaluations. Differences in study populations, prior exposure to the virus, or the timing of antibody measurements could also explain the varying findings across studies. These findings highlight the complexity of immune responses across different age groups and suggest further exploration of age-specific vaccination strategies.

Our analysis revealed that, after adjustment, factors such as sex, wealth index, and BMI were not significantly associated with antibody levels. This indicates that the COVID-19 vaccines used in this study elicit robust immune responses across diverse demographic groups, which is encouraging for public health strategies focused on achieving broad population coverage.

This study has several limitations that should be acknowledged. First, the cross-sectional design limits our ability to establish causality and track long-term antibody persistence. While our findings show a clear 15–16% monthly decline in antibody levels, we recognize that longitudinal studies are needed to better understand the temporal dynamics of immune responses. Antibody reduction is a dynamic process, and single time-point measurements may fail to capture the full trajectory of immunity. Moreover, our focus on antibody responses, while informative, offers only a partial perspective on immune protection. Future studies should incorporate assessments of cellular immunity to provide a more holistic understanding of vaccine-induced protection. Additionally, while key covariates were controlled for, unmeasured factors such as prior exposure to different SARS-CoV-2 variants, differences in immune memory, or individual immune profiles may have influenced the results. Furthermore, due to logistical constraints, this study focused on a specific urban population in Dhaka, which may limit the generalizability of the findings to other regions with different sociodemographic profiles. Future research should explore the effectiveness of booster strategies in more diverse settings, including rural areas and populations with limited access to healthcare.

## 5. Conclusions

This study offers valuable insights into antibody responses to homologous and heterologous COVID-19 booster vaccinations in Bangladesh. The observed decline in antibody levels over time highlights the need for continued monitoring and the consideration of additional booster doses to maintain protection. Further research with a robust study design is necessary to develop pragmatic vaccination strategies for Bangladesh and similar settings.

## Figures and Tables

**Figure 1 vaccines-13-00067-f001:**
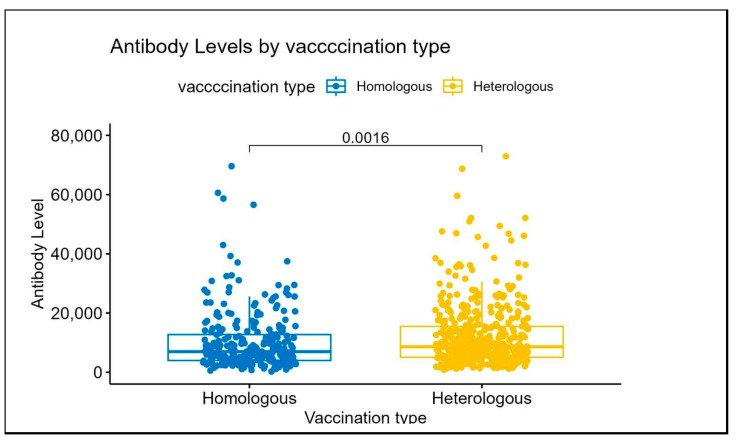
Antibody titer level by type of vaccination.

**Figure 2 vaccines-13-00067-f002:**
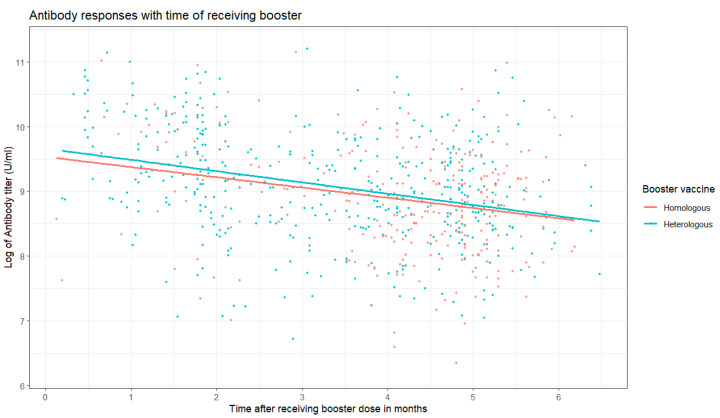
Antibody responses with time of receiving booster by homologous and heterologous group.

**Table 1 vaccines-13-00067-t001:** Sociodemographic and nutritional characteristics, health status, and antibody titers of the participants.

Characteristic	Homologous, *n* = 279	Heterologous, *n* = 444	Overall, *n* = 723
Age (years)	41.3 (±13.50)	36.8 (±11.85)	38.5 (±12.69)
Age (years)			
<30 years	60 (21.51%)	142 (31.98%)	202 (27.94%)
31–40 years	73 (26.16%)	148 (33.33%)	221 (30.57%)
>40 years	146 (52.33%)	154 (34.68%)	300 (41.49%)
Sex			
Male	89 (31.90%)	216 (48.65%)	305 (42.19%)
Female	190 (68.10%)	228 (51.35%)	418 (57.81%)
Level of education			
No education	34 (12.19%)	36 (8.11%)	70 (9.68%)
Primary completed	75 (26.88%)	73 (16.44%)	148 (20.47%)
Secondary completed	139 (49.82%)	247 (55.63%)	386 (53.39%)
Higher	31 (11.11%)	88 (19.82%)	119 (16.46%)
Occupation			
Service (govt./non-govt.)	77 (27.60%)	231 (52.03%)	308 (42.60%)
Housewife	134 (48.03%)	114 (25.68%)	248 (34.30%)
Day labor/unskilled worker	64 (22.94%)	89 (20.05%)	153 (21.16%)
Student	4 (1.43%)	10 (2.25%)	14 (1.94%)
Crowding index			
<2	63 (22.58%)	124 (27.93%)	187 (25.86%)
2 to 3	185 (66.31%)	279 (62.84%)	464 (64.18%)
>3	31 (11.11%)	41 (9.23%)	72 (9.96%)
Wealth index			
Poor	135 (48.39%)	177 (39.86%)	312 (43.15%)
Middle	121 (43.37%)	165 (37.16%)	286 (39.56%)
Rich	23 (8.24%)	102 (22.97%)	125 (17.29%)
Body Mass Index (BMI)			
Underweight (<18.5 kg/m^2^)	11 (3.94%)	23 (5.18%)	34 (4.70%)
Normal weight (18.5–24.9 kg/m^2^)	178 (63.80%)	252 (56.76%)	430 (59.47%)
Overweight (25–29.9 kg/m^2^)	75 (26.88%)	143 (32.21%)	218 (30.15%)
Obesity (≥30 kg/m^2^)	15 (5.38%)	26 (5.86%)	41 (5.67%)
Duration after receiving last booster dose in months	4.5 [3.60–5.10]	3.6 [1.88–4.78]	4.1 [2.14–4.90]
COVID-19 antibody in serum Anti-SARS-CoV2-S (U/mL)	6958.0 [3974.00–12,728.50]	8597.0 [5053.00–15,482.25]	7985.0 [4630.00–14,716.00]
Mean (±SD); *n* (%); Median [IQR]			

**Table 2 vaccines-13-00067-t002:** Morbidity and other health characteristics.

Known Morbidities Reported	Homologous, *n* = 279	Heterologous, *n* = 444	Overall, *n* = 723
(multiple responses)	*n* (%)	*n* (%)	*n* (%)
Diabetes	31 (11.11%)	28 (6.31%)	59 (8.16%)
Cancer	3 (1.08%)	5 (1.13%)	8 (1.11%)
Heart disease	15 (5.38%)	13 (2.93%)	28 (3.87%)
Respiratory disease	10 (3.58%)	14 (3.15%)	24 (3.32%)
Liver disease	1 (0.36%)	2 (0.45%)	3 (0.41%)
Kidney disease	4 (1.43%)	0 (0.00%)	4 (0.55%)
History of COVID-19 infection	10 (3.58%)	23 (5.18%)	33 (4.56%)
Number of times of COVID-19 infection			
No	269 (96.42%)	421 (94.82%)	690 (95.44%)
Once	9 (3.23%)	17 (3.83%)	26 (3.60%)
Twice and more	1 (0.36%)	6 (1.35%)	7 (0.97%)
Mobility	38 (13.62%)	50 (11.26%)	88 (12.17%)
Self-care	103 (36.92%)	200 (45.05%)	303 (41.91%)
Usual work	104 (37.28%)	218 (49.10%)	322 (44.54%)
Pain	82 (29.39%)	210 (47.30%)	292 (40.39%)
Anxiety	31 (11.11%)	41 (9.23%)	72 (9.96%)
Difficulty remembering	78 (27.96%)	130 (29.28%)	208 (28.77%)
Feeling weak	89 (31.90%)	137 (30.86%)	226 (31.26%)
Difficulty concentrating	23 (8.24%)	28 (6.31%)	51 (7.05%)
Change in appetite	16 (5.73%)	20 (4.50%)	36 (4.98%)

**Table 3 vaccines-13-00067-t003:** Factors affecting COVID-19 antibody level among the participants (*n* = 723).

	Unadjusted		Adjusted	
Characteristic	Exponential Coefficient (95% CI)	*p*-Value	Exponential Coefficient (95% CI)	*p*-Value
Type of vaccination				
Homologous	—		—	
Heterologous	1.23 (1.08, 1.40)	0.001	1.10 (0.97, 1.25)	0.146
Duration after receiving last booster dose in months	0.84 (0.81, 0.87)	<0.001	0.85 (0.81, 0.88)	<0.001
Age (years)				
<30 years	—		—	
31–40 years	1.15 (0.98, 1.35)	0.091	1.21 (1.03, 1.41)	0.020
>40 years	1.18 (1.01, 1.37)	0.035	1.23 (1.07, 1.43)	0.005
Sex				
Male	—		—	
Female	0.80 (0.70, 0.90)	<0.001	0.90 (0.80, 1.02)	0.099
Wealth index				
Poor	—		—	
Middle	1.18 (1.03, 1.35)	0.019	1.09 (0.95, 1.24)	0.224
Rich	1.16 (0.97, 1.38)	0.101	0.92 (0.77, 1.10)	0.358
BMI				
Normal (18.5–24.9 kg/m^2^)	—		—	
Underweight (<18.5 kg/m^2^)	1.25 (0.93, 1.68)	0.148	1.12 (0.85, 1.49)	0.417
Overweight (25–29.9 kg/m^2^)	1.17 (1.02, 1.35)	0.025	1.10 (0.96, 1.25)	0.179
Obesity (≥30 kg/m^2^)	1.23 (0.94, 1.62)	0.134	1.16 (0.89, 1.51)	0.258
History of COVID-19 infection				
No	—		—	
Yes	1.13 (0.84, 1.52)	0.420	1.00 (0.75, 1.34)	0.989

## Data Availability

Due to icddr,b’s data access policy regarding participant-identifying information, the data are available upon request from the Research & Clinical Administration and Strategy (RCAS) at icddr,b. Researchers who meet the criteria for access to confidential data can request access via the following link: icddr,b data policies.

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
