# Peer review of "Comparing Antibody Responses to Homologous vs. Heterologous COVID-19 Vaccination: A Cross-Sectional Analysis in an Urban Bangladeshi Population"

_vaccines, 2025, doi:10.3390/vaccines13010067_

Round 1

Reviewer 1 Report

Comments and Suggestions for Authors

This study compares inbreeding and heterogeneous booster strategies among urban residents of Bangladesh. 723 participants were recruited from a diverse urban population in Dhaka, and the approach using blood sample analysis and structured questionnaires provides reliable data.

However, as discussed in the text, it is not possible to assess the persistence of the immune response and to track antibody reduction over time.

Even after adjusting for key factors such as age, gender, and BMI, other confounding factors (e.g., history of asymptomatic infection or individual immune profile) are not adequately taken into account.

The persistence of the immune response cannot be assessed and antibody decline cannot be tracked over time. Antibody reduction is a dynamic process, and measurements at a single point in time may not provide a complete picture of immunity.

It has not been shown to correlate with clinical outcomes such as infection rates or risk of severe disease. Antibody titers are only part of the immune response and lack assessment of cellular immunity.

Factors such as age, comorbidities, and history of infection need to be analyzed in more detail. It is known that antibody titers increase markedly with hybrid immunity, markedly when there is a history of infection in the past.

Please try to mention the above mentioned points in detail.

Author Response

This study compares inbreeding and heterogeneous booster strategies among urban residents of Bangladesh. 723 participants were recruited from a diverse urban population in Dhaka, and the approach using blood sample analysis and structured questionnaires provides reliable data. However, as discussed in the text, it is not possible to assess the persistence of the immune response and to track antibody reduction over time. Even after adjusting for key factors such as age, gender, and BMI, other confounding factors (e.g., history of asymptomatic infection or individual immune profile) are not adequately taken into account.

Response: Thank you. We agree with your comment and have included this point as part of the study's limitations. Regarding the history of asymptomatic infections, respondents were unable to provide adequate information, which prevented us from incorporating it into the analysis.

The persistence of the immune response cannot be assessed and antibody decline cannot be tracked over time. Antibody reduction is a dynamic process, and measurements at a single point in time may not provide a complete picture of immunity.

Response: Thank you. We agree with your comment and added it to the limitations section of this study. 

It has not been shown to correlate with clinical outcomes such as infection rates or risk of severe disease. Antibody titers are only part of the immune response and lack assessment of cellular immunity.

Response: We have added text acknowledging that: "Our focus on antibody responses, while informative, provides only a partial view of immune protection. Future studies should include assessment of cellular immunity to provide a more comprehensive understanding of vaccine-induced protection."

Factors such as age, comorbidities, and history of infection need to be analyzed in more detail. It is known that antibody titers increase markedly with hybrid immunity, markedly when there is a history of infection in the past.

Response: You make an excellent point, and we completely agree. In this study, the majority of respondents (~96%) reported no history of infection. We assume that many were unable to identify or confirm an infection due to a lack of proper diagnostic investigations. This likely led to significant underreporting of infection history, which may explain why it showed no association during statistical analysis. 

Reviewer 2 Report

Comments and Suggestions for Authors

Kazi Istiaque Sanin et al compare in their study the antibody response of the urban Bangladeshi population to homologous and heterologous immunization. 

Their result is that in agreement with other studies the antibody levels of patients undergoing a heterologous immunisation, meaning immunisation with two different vaccines, are slightly higher than when receiving a homologous immunisation. However the higher level of antibodies diminishes more rapidly than the antibody level from a homologous immunisation. 

The result of this article is in so far interesting as the differences between the two immunisation schemes are minor in comparison with the inter-individual variations (see figures 1 and 2). The data suggest that it really does not matter whether a particular patient receives a heterologous or a homologous immunisation because other factors not covered in this investigation are much for significant for the variation in inter-individual antigen response. 

I think the authors should focus more on this aspect of the results.

With the conclusions the authors propose - a simple confirmation of data already observed by other studies - I think that the article does not contribute enough new information to justify publication. 

Author Response

Response letter

Comments and Suggestions from the second reviewer

Their result is that in agreement with other studies the antibody levels of patients undergoing a heterologous immunisation, meaning immunisation with two different vaccines, are slightly higher than when receiving a homologous immunisation. However the higher level of antibodies diminishes more rapidly than the antibody level from a homologous immunisation. 

The result of this article is in so far interesting as the differences between the two immunisation schemes are minor in comparison with the inter-individual variations (see figures 1 and 2). The data suggest that it really does not matter whether a particular patient receives a heterologous or a homologous immunisation because other factors not covered in this investigation are much for significant for the variation in inter-individual antigen response. I think the authors should focus more on this aspect of the results.

Response: Thank you. We agree with your thoughtful comment. Many other factors are not covered in this investigation, and we have included them as limitations of our study. 

With the conclusions the authors propose - a simple confirmation of data already observed by other studies - I think that the article does not contribute enough new information to justify publication. 

Response: Thank you. While our findings align with prior research in some areas and strengthen existing evidence, we believe our study offers few unique contributions:

  • It is the first to explore these patterns within the distinctive demographic and healthcare context of Bangladesh.
  • The study incorporates a comprehensive analysis of sociodemographic factors in a densely populated urban setting.
  • Its findings have implications for optimizing vaccination strategies in resource-limited settings.

Reviewer 3 Report

Comments and Suggestions for Authors

Dear Authors,

The Authors have taken up an important topic from an epidemiological perspective related to the effectiveness of vaccinations against Covid-19.

In general, I consider the article to be quite good, but the authors should include a few comments, which are below.

Introduction – due to the fact that the journal Vaccines has an international reach, it will be interesting for readers to also refer to the situation in other Asian countries, but also in the USA, Europe and Africa.

Methodology – The authors should write whether the study has been approved by the bioethics committee and provide the number of this approval.

Apart from that, I have no more comments.

Author Response

Response letter

Comments and Suggestions from the third reviewer

In general, I consider the article to be quite good, but the authors should include a few comments, which are below.

Introduction – due to the fact that the journal Vaccines has an international reach, it will be interesting for readers to also refer to the situation in other Asian countries, but also in the USA, Europe and Africa.

Response: Thank you. We have revised the introduction to incorporate your suggestion.

Methodology – The authors should write whether the study has been approved by the bioethics committee and provide the number of this approval.

Response: We have included the details of the bioethics committee with approval number in the “Institutional Review Board Statement” section of the manuscript

Round 2

Reviewer 1 Report

Comments and Suggestions for Authors

The authors have adequately answered my questions.